# Lived Experiences of Suicide Bereavement within Families: A Qualitative Study

**DOI:** 10.3390/ijerph192013070

**Published:** 2022-10-11

**Authors:** Clémence Creuzé, Laurène Lestienne, Maxime Vieux, Benoit Chalancon, Emmanuel Poulet, Edouard Leaune

**Affiliations:** 1Centre Hospitalier Le Vinatier, 95 Boulevard Pinel, 69678 Bron, France; 2INSERM, U1028, CNRS, UMR 5292, Lyon Neuroscience Research Center, PSYR2 Team, 69000 Lyon, France; 3Hospices Civils de Lyon, 69002 Lyon, France

**Keywords:** suicide, bereavement, family, lived experiences, qualitative

## Abstract

Backround: The lifetime prevalence of suicide exposure in the family is estimated at 3.8% in the general population. Familial bonds can constitute a crucial factor in determining individual bereavement outcomes via the interactional and communicational aspects of the bereavement process within the family. However, the literature on the lived experiences of suicide bereavement within the family remains scarce. Method: Our objectives were to qualitatively (a) assess the impact of suicide on different types of family members, (b) evaluate the interactions between the familial and individual bereavement processes, and (c) obtain precise insights into the familial interactions that occur following a suicide. We performed a qualitative study by conducting semidirected interviews with family members who had been bereaved by suicide. Computer-based and manual thematic analyses were used for data analysis. In order to mitigate biases related to the qualitative design of the data collection, two main measures were undertaken, namely triangulation and saturation. Results: Sixteen family members bereaved by suicide participated in our study, including a majority of women (*n* = 12). Mean age of participants was 56.4 and mean duration of suicide bereavement was 10.5 years. Most of the relatives deceased by suicide were middle-aged men who died by hanging or firearm. A total of six themes emerged from the analyses, namely (1) “familial trauma”, (2) “external adversity”, (3) “individual bereavement and familial interactions”, (4) “communicational and relational processes within the family”, (5) “perceived help and support within the family” and (6) “evolution over time”. Conclusion: We reported that suicide bereavement significantly impacts internal familial interactions via complex emotional and communication processes. Family conflicts, taboos or cohesion can occur in the aftermath of a suicide. We also found that familial coping strategies can shape individual reactions to the death on the part of each member of the family.

## 1. Introduction

Every occurrence of suicide affects several family members of the deceased at both an individual and a collective level. The psychologist Edwin Shneidman was the first to estimate the number of impacted relatives as six based on his daily clinical practice [1], but recent epidemiological studies have provided more precise estimations [2,3]. Berman [2], for example, found that between five and fifteen family members are affected by one instance of suicide in the family. According to a recent meta-analysis [3], the lifetime prevalence of suicide exposure in the family is estimated to be 3.8% in the general population, and so nearly one person in 25 will be exposed to a suicide in his or her family.

Suicide bereavement is known to be associated with numerous negative outcomes, including impacts on both mental health (depression, suicidal behaviors, anxiety, posttraumatic stress disorder, and prolonged grief) and physical health (chronic pain, cardiovascular diseases, etc.) [4]. Psychosocial outcomes include social and work withdrawal due to the stigma and emotional impact of the suicide bereavement process [5]. Guilt, self-blame and shame are particularly frequent among people who have been bereaved by suicide, impeding their ability to navigate through the social world [4]. Social support is known to be a strong predictor of posttraumatic growth in people who have been bereaved by suicide, thus highlighting the need to promote social bonds among this at-risk population [6,7]. Familial bonds can thus constitute a crucial factor in determining individual bereavement outcomes via the interactional and communicational aspects of the bereavement process within the family [8].

However, the literature concerning the impact of suicide bereavement in the family remains scarce [8]. Only a few studies of low quality have assessed the impact of suicide on family members [8], and most published studies have examined only one type of suicide survivor (i.e., parent, child, partner) or included both people who have been bereaved by suicide and those who have been bereaved by other violent causes of death [9]. In a longitudinal study concerning the impact of violent death among adolescents aged between 12 and 18, for example, Lohan and Murphy [9] assessed the perceptions of family functioning exhibited by parents in the aftermath of the death. The observed difficulties notably included decreases in family cohesion and adaptation but did not include any differences between families that had been bereaved by suicide and those that experienced other causes of death. According to these authors, parental functioning may influence surviving family members, especially children, who may be faced with parents who are less emotionally available [9]. In their qualitative study of widows, McNiel, Hatcher, and Reubin [10] found that women whose husbands had died by suicide experienced more guilt and blame from their families than those who had lost a husband in an accident. A better understanding of the impact of suicide on the emotional, communicational and interactional processes that occur within a family is thus necessary. In 2008, Cerel, Jordan and Duberstein [8] recommended that both qualitative and quantitative studies be conducted to examine the impact of suicide bereavement on the family. However, no well-designed and high-quality qualitative study has been conducted since this call to action.

Understanding the impact of suicide bereavement within families is crucial as familial bonds constitute the first social support for many people bereaved by suicide [8]. The lack of familial support could therefore have significant and critical impacts on family members, especially regarding the negative psychological and psychiatric outcomes previously mentioned. Thus, assessing how the communicational and relational processes within the family shape individual bereavement could offer important cues to prevent negative outcomes and favor posttraumatic growth in family members. In the same time, individual processes and negative outcomes in members of the family may also shape the interactions within the family by increasing overall familial burden. Results from studies on this issue could thus inform how to support families and their members in coping with both individual and familial processes of suicide bereavement.

Our objectives were thus to qualitatively (a) assess the impact of suicide on different types of family members, (b) evaluate the interactions between the familial and individual bereavement processes, and (c) obtain precise insights into the familial interactions that occur following a suicide.

## 2. Method

We performed a qualitative study by conducting semidirected interviews with family members who had been bereaved by suicide.

### 2.1. Recruitment of Participants

The participants were recruited between September 2020 and January 2021 in three ways: via social media, through charities dedicated to people who have been bereaved and by way of the Center for Suicide Prevention associated with the Centre Hospitalier le Vinatier (Lyon, France). Calls to participate were sent via social media and email to putative participants. The inclusion criteria were as follows: people who had been bereaved by suicide for more than one year, who had lost a nuclear family member (i.e., a child, sibling, parent, wife or husband, or partner) and who spoke fluent French. People under 18, people who had been bereaved by causes other than suicide and people who had been bereaved for less than one year were excluded from the study.

### 2.2. Data Collection

The semidirected interviews were conducted at the Center for Suicide Prevention by the first author. Due to the COVID-19 pandemic, the interviews could be performed face-to-face, by phone or by videoconference.

The following sociodemographic characteristics of participants were collected by the interviewer at the beginning of the interviews: age, gender, relation to the deceased and duration of suicide bereavement. Data concerning the relative who had committed suicide were also collected: gender, age at death and means of suicide. The structure of the family was also collected (nuclear family, separated family or stepfamily). The semi-structured interviews lasted between 60 and 90 min each and were conducted by the first author. Each interview started with the following sentence: “How did your family experience suicide bereavement?” followed by reminders or reformulations when the discussion strayed too far from the research subject or began to fade. The participants were interviewed regarding their individual experiences of the impact of the death by suicide of their relative on their families, including their perceptions of bereavement on their own part and on the part of their relatives and the ways in which the impact of suicide bereavement on their family evolved throughout the years. The content of the interview was recorded and anonymized.

### 2.3. Data Analysis

Based on the narratives uncovered through the semidirected interviews, a thematic analysis was conducted over five chronological phases as discussed by Sibeoni et al. [11]: (1) reading, (2) descriptive coding, (3) conceptual coding, (4) identifying themes and (5) producing one coherent thematic structure (Table 1).

In accordance with the method recently developed by Renz et al. [12], two data analyses (i.e., manual and computer-based analyses) were combined to enhance the reliability of the results. The first method is based on a manual thematic analysis [13]. The authors examined the apparent messages through a repeated reading of the transcripts to achieve immersion and obtain a sense of the whole. In addition, this initial reading allowed us to define thematic and formal categories that would be relevant to later coding of the speech. Units of meaning were then independently identified, categorized and related to one another to identify axes of transversal meanings. This process allowed us to classify the elements and produce a simplified representation of the raw data. The computer-based thematic analysis using NVivo software was performed by the first author. NVivo is a computer-assisted qualitative data analysis software that allows for qualitative inquiry beyond the level of coding, sorting and retrieving the data [14]. The benefits of using NVivo pertain to the fact that it allows teams of researchers to systematically and rigorously synthesize qualitative data.

### 2.4. Mitigation of Bias

To mitigate bias related to the qualitative design of the data collection, two main measures were employed, namely, triangulation and saturation.

*Triangulation* refers to the use of multiple methods or data sources in qualitative research to develop a comprehensive understanding of phenomena [15]. Denzin [16] and Patton [15] identified four types of triangulation: (a) method triangulation; (b) investigator triangulation; (c) theory triangulation; and (d) data source triangulation. Two types of triangulation were particularly employed in our study. First, the analysis process was submitted to triangulation; the analysis was summarized and subsequently discussed between the authors. Second, investigator and theoretical triangulation were also employed by ensuring the involvement of researchers from several disciplines (e.g., social psychology, psychiatry, public health, nursing and health philosophy).

*Saturation* is a methodological means of ensuring the reliability and representativeness of data collected via a qualitative approach by continuing the data collection process until new data no longer contribute new insights to the outcome of the study [17]. According to the taxonomy developed by Saunders et al. (2018) [17], the following four types of saturation are defined: (a) theoretical saturation (i.e., the development of theoretical categories), (b) inductive thematic saturation (i.e., the emergence of new codes or themes), (c) a priori thematic saturation (i.e., the degree to which identified codes or themes are exemplified in the data), and (d) data saturation (i.e., the degree to which new data repeat what was expressed in previously collected data). Given the homogeneity of the sample (i.e., French family members bereaved by suicide), a total of 15 to 20 participants were expected to be necessary to reach the point of saturation. The study was concluded after 16 participants had been included, as the four types of saturation were reached. Indeed, no new codes nor new themes emerged during the final two interviews (theoretical and inductive thematic saturation) and the data collected during these two interviews repeated what was previously reported by the other participants (data saturation). Moreover, a priori thematic saturation was also reached as the codes and themes that we identified were highly exemplified in the data collected during all the interviews.

### 2.5. Ethical Approval

The study received ethical approval from the Ethics Committee of the Claude Bernard Lyon 1 University in July 2020 (n° IRB 2020-07-07-03).

## 3. Results

### 3.1. Participants

A total of 16 family members who had been bereaved by suicide participated in our study. Participants were recruited via social media (37.5%), charities (31.25%) and the Center for Suicide Prevention (31.25%). Most participants were women (75%), and they had been bereaved by the death of a child (50%), a loved one (18.75%) or a sibling (18.75%). The mean age of participants was 56.4 years, and the mean duration of suicide bereavement was 10.5 years. Most of the deceased were men who had died by hanging or the use of firearms. The characteristics of participants and their relatives who had died by suicide are listed in Table 2.

### 3.2. Themes

A total of six themes emerged from the analyses, namely, (1) “familial trauma”, (2) “external adversity”, (3) “individual bereavement and familial interactions”, (4) “communicational and relational processes within the family”, (5) “perceived help and support within the family” and (6) “evolution over time”. These themes and their subthemes are described below and explicated by reference to quotations from the participants. The numbers of participants supporting each theme are given in Table 3.

#### 3.2.1. Familial Trauma

Terms such as “tsunami”, “cataclysm”, “conflagration” or “bomb” were used to describe the impact of suicide on family members. Most participants highlighted the violence and the brutality of the event for their families.


*“The family experienced it as an explosion […]. At the level of our nuclear family, we came close to disaster for several months under the explosion, the tsunami. It was so violent.”*
(P6)


*“Obviously, it was a shock, and the shock, it is as if as soon as his parents knew it, it impacted us all, the family, with the same shock wave.”*
(P2)

The family experienced a state of shock and paralysis during the initial days and weeks following the suicide, which occasionally persisted for months.


*“The conflagration is really very hard and extremely painful. It lasted between six months and a year. Afterward, in our family, this is what happened. For six months, everybody was shocked and traumatized. We had the impression of being a little weightless; we’re no longer in the real world, we’re in a weird world, and it was true for the whole family.”*
(P6)

Family balance was also described as being impeded by the suicide due to the void left by the deceased.


*“I took pictures of the four of us for a long time, the remaining four girls, and I always tried to put my sisters in the same order, the eldest, the middle…. Every time I look at these photos, there’s no physical hole, but for me, something is missing.”*
(P1)


*“Afterward, it created a void, an imbalance that we compensated for, so I don’t know whether to say like that, how shall I put it, this void, this brutal absence and this violence”*
(P11)

#### 3.2.2. External Adversity

Several factors pertaining to experiences of adversity by family members were reported, including with respect to the police inquiry, the management of logistics following the death, material difficulties and social stigma. The police inquiry led to feelings of guilt and suspicion for some families, especially when they were insufficiently informed of the relevant administrative procedures.


*“My father and I went to the police station together, where we started to be asked very targeted questions: ‘What did your brother do? What job? What knowledge did he have? What types of dating? Do you think there are people who were mad at him? Do you know if he was involved in any shady things?’”*
(P1)

Managing logistics during the aftermath of the death was a critical challenge for families, and it involved the individual competences of several members of the family.


*“My father had planned everything. He had left us all his codes and all his files, so that we wouldn’t look everywhere in his papers. He had everything listed in his files, so we only had to execute things.”*
(P10)

Financial difficulties were also reported, notably by families who had been bereaved by the suicide of their father.


*“There was another problem; it was a financial problem for the family because there was no more income at home.”*
(P16)

Struggling to obtain financial compensation and manage administrative procedures was described as particularly challenging.


*“I had to justify and show violence to obtain the papers, to get everything I was supposed to get from my husband’s business. I had to hire a lawyer, all the same, in all these stories, in all these facts that I have just told you.”*
(P11)

Social stigma was also particularly difficult to face for the families, mostly in the form of autostigma.


*“I think we are stigmatized in the minds of everyone who kindly came to us; we are stigmatized in the sense that ‘in this family, there was a suicide.’ I could feel it.”*
(P15)

#### 3.2.3. Familial Interactions and Individual Bereavement

Participants reported the major impact of bereavement on familial interactions, including the associated emotional and relational processes. The participants notably reported discrepancies in the individual reactions and coping strategies of the family members, which were reported in terms of several dimensions: emotional responses, symptoms of grief, explanations of the causes of the suicide and coping strategies.


*“There were four of us; we were still four girls, well, the four of us, we had an explanation that was only true for us. My sisters, they probably have an explanation for my brother’s death, but we can’t confront it, because anyway if I tell them what I think, they’ll tell me that I’m talking nonsense, because it’s not their construction; plus, we all have professional biases”*
(P1)

In some families, these discrepancies were respected by all members of the family, while in other families, such differences could lead to conflicts and mutual misunderstandings. The social and emotional support provided by family members to one another could also be hampered by these differences in ways of coping with the death.


*“After the funeral, everyone reacted a little differently. Some got help from a psychologist to deal with the shock, others didn’t. And then also, for example, I didn’t feel anger while in my close circle; my mother and my sister, they were angry.”*
(P10)

Couples were particularly vulnerable to coping discrepancies and a loss of mutual support that they had previously provided to one another.


*“About six months after his death, I said to myself ‘we won’t be able to stay together,’ and I said to myself ‘we will have to separate.’ Not because we don’t love each other but because the suffering of the other is deep, I felt that; I said to myself “my wife is taking me to a well without a bottom” and “if I want to survive, we will have to divorce.’”*
(P6)

A “climate of blame” was reported in some families, indicating a situation in which one of several members of the family was considered to be guilty regarding the death.


*“My mother was very aggressive with him, and indeed I knew that he felt bad at the end of his life and that she had still been very aggressive with him lately. The feeling that she had killed him… and also the feeling, at some point there, it came back a little bit, but for years, I have felt that the day I lost my brother, I also lost my mother because that’s something I couldn’t get over, the attitude she had with him.”*
(P4)

This situation was particularly reported in the case of widows who were considered to be guilty for the death by their stepfamilies, leading to conflicts and the dissolution of relationships.


*“So afterward, in terms of relations, with my daughter-in-law, things are better because time has passed, but deep inside me there is… I saw her once or twice; we talked a little, and she cried, telling her that indeed I held her responsible for the death of E… I hold her responsible for the harm she may have done him by leaving him, by coming back”*
(P5)


*“On the side of my ex, of my husband’s family, it became my ex-family; that is to say, in fact, it was a tear, a split. It was extremely violent because for them, I am part of the person who did not… who is guilty, to say the word.”*
(P11)

A “climate of fear” was also reported by the majority of the participants. The fears reported included fear regarding the death of another member of the family, fear of the heritability of suicidal behaviors and fear pertaining to building new relationships. Fear of another suicide was notably highly prevalent, leading to the adoption of protection strategies regarding the members of the family who seemed to be the most vulnerable and/or impacted by the death. This fear regarding subsequent suicide could persist for decades. Building a new relationship was reported as being very difficult, as the fear of loss significantly impeded individuals’ ability to create close relationships.


*“In fact, I found myself pregnant and then I had a panic attack, a crisis; here I began to imagine that I risked having a child who looked like my brother. So, there it was a nightmare, I was very scared; I was so scared that in fact the pregnancy did not last.”*
(P3)


*“When we say there’s nothing worse than losing a child, I’m not telling you the impact on the ability to become a parent yourself. I think it’s a message, it’s a bit heavy; besides, I don’t have children. It’s not the only reason, but I think it’s part of the reason. Then, even beyond that, I would say to attach myself to someone is to run the risk of a loss, as immeasurable. I have never succeeded.”*
(P4)


*“It was like a sword of Damocles over their heads, you see; it was really something (talk about a suicidal act on his part following his son’s suicide) that they were afraid would occur.”*
(P2)

#### 3.2.4. Communicational and Relational Processes within the Family

The death of a member of the family by suicide can have significant consequences for the communicational and relational processes that occur within the family. Discussing the death of their relative by suicide was reported as a great challenge in some families, potentially leading to a taboo and silence over a period of months or years.


*“For me, for a long, long time, for years, Christmas meals were hell because there was silence. We didn’t talk anymore, even though we were a family that talked a lot, we argued, we bickered, we sang… Silence, it was terrible, terrible, terrible.”*
(P1)

Expressions of emotion, particularly sadness, guilt or anger, on the part of some members of the family can hamper internal family discussions, as other members of the family prefer not to accentuate the painful feelings expressed by those who were most impacted. Family secrets can occur as a consequence of the taboo or due to the fear of explaining the cause of the death to children in the family.


*“It remained a taboo for a very long time, I think we experienced; I don’t know if you call that a complex mourning, a traumatic mourning, there are now words to say that (…) We were in such pain during those years that indeed it was not sayable.”*
(P4)

Sharing emotional experiences related to the death such as turmoil or emotional distress was described as a difficult task. Providing and receiving support from the family may thus become difficult as some of the members of the family do not feel comfortable expressing their feelings.


*“They don’t want to hurt their spouse by showing it, that’s an inextricable problem, you don’t want to show your spouse, one way or the other, you don’t want to show to your partner that you’re upset. I was going upstairs to the bathroom and then started crying quietly because I knew my husband was downstairs watching TV.”*
(P15)

Interactional processes can be shaped in different ways by the death. Most participants reported that suicide bereavement accentuated preexisting interactions within the family, especially when those interactions were already complicated.


*“The relationship with my mother was already very complex before, and in fact, that was the extra thing that made it even more... Even more knotted, complicated... Painful, but in fact we can’t say that it’s a trigger; it’s part of an overall dynamic, to which this fact and his attitude before this fact largely contributed, and after that too.”*
(P4)

Some participants reported critical and significant changes in interactions among the members of the family, including both negative (conflicts, family estrangement) and positive (family reunification, familial support) shifts.


*“We had been together for twenty-three years, so I had reconstituted with my husband, a family of, one side, him with his children, me with mine, and the grandchildren of all this little world, and we were all united, we had reconstituted a cocoon of agreement following this death […] I don’t see my husband’s children anymore, I don’t see my grandchildren anymore, that’s sad. […] They don’t call me anymore, and we don’t see each other anymore. The link is broken when I thought it was strong, but no, finally, there’s nothing left, that’s it.”*
(P9)


*“A rapprochement, yes, of my children compared to before. And it’s true that sometimes, for example, before, they hardly ever called me on the phone. Yesterday, for example, S. called me, and I said to myself ‘oh, that’s great.’ It seems crazy, but a phone call from my children made me very happy, because before they never called me.”*
(P13)

Parents or children who have been bereaved by suicide highlighted the difficulties in managing parental roles after the death (supporting children, providing affective reinsurance, managing day-to-day details), so that some children can assume responsibility as parents or adults and thus reverse parent–child interactions.


*“I said to myself ‘I am no longer capable of educating my three other children,’ so it was even harder. What to do? What must we not do? What must we say? What must we not say? And so, we revisited everything that in fifteen years we should perhaps have done or not done. We no longer knew how to educate our children. Should we overprotect them? Should we, on the contrary, leave them completely free? So, all the educational benchmarks were shattered for quite a while; I think at the family level, we no longer knew what to do.”*
(P6)


*“Your parents, they are not available for you because they think about the one who is no longer there (…) In addition to the one who is no longer there, they idealize him. So, how do you do exist? Who are you, what are you for? And it goes further, because in fact there is also the fact that maybe it would have been better if it were you who died.”*
(P4)

Parents who have lost a child by suicide notably doubt their ability to care for their remaining children. This doubt can also lead to a silencing of the grieving process in children who do not want to worry their parents. The “silent bereavement of siblings” has been reported by several participants who were bereaved by the suicide of a brother or sister.


*“It’s also good that you’re going to take care of your parents because it’s them who are suffering, it’s not you, so in fact the relationship, well, the relationship is completely reversed (…) Your parents, they suffer so much that you shut up.”*
(P4)


*“We often forget siblings in suicides; we often think of parents, and we often forget brothers and sisters.”*
(P7)

#### 3.2.5. Perceived Help and Support within the Family

As reported by the participants, the family itself can constitute an efficient coping strategy through the provision of emotional and practical support to its members.


*“Already, we survived, we’re still together. U. is there too, so we’re very close. There was a lot of love at home for four, and in fact the love for four has been divided into three, so you see it was at least love at home, it was at that level, we did very well.”*
(P7)

Sharing information concerning suicide and suicide bereavement within the family and sharing spiritual beliefs or common projects were particularly helpful.


*“A piece of advice that I have given to a lot of people, it’s to talk about the person who has left, that they continue to have a place in the family.”*
(P6)


*“My son came with me, and when we were in the car we had signs, we had signs, that is to say that in the car, the music changed on its own. I like classical music, so it put me in classical music, and it didn’t stop. We could see that there was an energy that ended. So, with A., we shared it together.”*
(P13)


*“For me, what was important was to rebuild both of us, to continue to live, to live differently, to continue to live, and also to continue to live for S.. Not to impose on him that I lost my sister and then my parents let themselves die.”*
(P8)

In contrast, the family can be perceived as unhelpful when conflicts or estrangement occur following the death.


*“I think that it is still today the resentment that they have towards each other, and we will say five years later, no, ten after, they broke off all relations on the pretext of an estrangement which was the resale of our family home.”*
(P12)

External social support was also identified as very critical. Circles of friends or professionals, health professionals or peers who have been bereaved by suicide were particularly identified as sources of efficient and helpful support for the family and its members, especially during the early stages of the bereavement process.


*“With regard to the two families, we did not really find support as we expected, so we really turned to friends, who knew and who were able to be present, with whom we were able to establish relationships.”*
(P11)


*“We were lucky enough to have immediate access to psychological support through the psychological crisis unit that had been set up by my company, and the human resources department opened up to me, opened up this possibility there at all, so we could call 24 h a day.”*
(P8)


*“I think that counselling, as I had very quickly, is a support, and fortunately it was there, because indeed I do not think that we can get out of it alone, in fact.”*
(P14)

However, a large majority of participants reported a perceived lack of support for their family following the death. In particular, they noted that they received no information concerning the existing resources and thus had to discover those resources by themselves.


*“I tried to look for associations of bereaved parents, but there is no end to suicide here, near where I live there is nothing, absolutely nothing”*
(P5)

The lack of knowledge of existing resources on the part of general practitioners was especially highlighted.


*“The first reflex we have when, unfortunately, it happens is to go see our GP, and I have the impression that GPs don’t even know it exists.”*
(P13)

The participants highlighted the need for early access to support and resources for families in the aftermath of the death.


*“You should receive a brochure by email or I don’t know anything about it, with all the useful phone numbers, all the existing charities, everything that is possible, and then a small booklet on mourning at the beginning, what are the reactions. There is something missing”*
(P6)

#### 3.2.6. Evolution over Time

The participants reported the global improvement of family members over time, with posttraumatic growth occurring among members of their families. This growth led to the opportunity for families to restore balance to the family, including improved communication among family members or the restoration of parent-child interactions.


*“I would say, overall, that it was an event in our lives that broke us, and like everything that doesn’t kill us, it strengthened us and then made us go down other paths.”*
(P8)


*“Today, we can have family meals, simple, peaceful family meetings; we can laugh, we can bicker, we can argue like a family, we can put things into words.”*
(P1)

For some families, despite its painful effects, the suicide was even reported to be an experience of growth for all the members of the family. Discovering unexpected individual and familial resources, developing deep and strong internal family bonds and improving mutual care among family members were, for example, reported as positive long-term effects within the family.


*“I know that in the couple, in the family, we have all developed abilities, springs, resources that we did not know, that we did not suspect.”*
(P6)


*“It’s positive for my relationship with my family, really, we tend to talk more when we didn’t talk a lot, even of things that hurt. We always talked about some things and others, but real things that hurt, we’ve always had a kind of modesty like that, it made it possible to say things better.”*
(P2)

## 4. Discussion

### 4.1. Summary of Results

To the best of our knowledge, we performed the first well-designed qualitative study to assess the impact of suicide bereavement on families and family members. We reached saturation after including a total of 16 people who had been bereaved by suicide and identified six main themes pertaining to the impact of suicide bereavement on the family, namely, “familial trauma”, “external adversity”, “individual bereavement and familial interactions”, “communicational and relational processes within the family”, “perceived help and support within the family” and “evolution over time”.

### 4.2. Interpretation of Results

According to our results, suicide bereavement can have significant and specific effects on families. A climate of guilt, taboo or stigma can, for example, be considered common impacts of suicide bereavement on family members. In a cross-sectional study on people who had been bereaved by suicide [18], 30% of participants reported a climate of blame or conflicts within their families. Moreover, in several studies, people who have been bereaved by suicide reported more conflicts and blame than did those who had been bereaved by other causes of accidental death [10,19,20]. Feelings of stigma are also known to be highly prevalent in people who have been bereaved by suicide and can negatively affect their psychosocial and mental health [5,21]. Our results demonstrate that feelings of shame, judgment and stigma can also impact the family as a whole. Thus, familial interventions targeting specific aspects of suicide bereavement should be implemented and evaluated. While no study has previously assessed the impact of familial grief interventions on people and families who have been bereaved by suicide, the *Family Bereavement Program* has been shown to be effective in mitigating the mental health and psychosocial outcomes of spousally bereaved parents [22] and parentally bereaved adolescents [23]. The *Family Bereavement Program* is a 12-session group program aimed at parentally bereaved children and adolescents and their caregivers that focuses on emotional and communication skills [24].

Interestingly, the interactions between individual and familial reactions to death and the bereavement process were particularly highlighted by the participants in this study. A lack of support within the family or deleterious interactions among the members of the family can negatively affect people who have been bereaved by suicide. Discrepancies in emotional and communicational coping strategies were particularly reported by the participants. In contrast, family can constitute a positive and strong source of support that can allow people to face bereavement processes [25]. Thus, clinicians should carefully assess familial interactions to effectively evaluate and support their patients who have been bereaved by suicide. Moreover, the way in which familial interactions shape individual reactions during the process of bereavement must be more clearly understood and widely investigated in future research, including through the use of surveys and cohorts featuring larger samples.

The perceived lack of support reported by the majority of participants suggests critical considerations for the support provided to families who have been bereaved by suicide. The putative roles of first responders (police, firemen, etc.), general practitioners and peers must be mentioned as notable, as these individuals can provide early access to help and support for families and family members [26]. Our results highlight both of the negative external interactions experienced by family members following the suicide of a relative. Notably, police inquiry can impact bereaved people and families negatively and should be targeted to improve the support provided to families following the death of a relative [27]. Online resources can also provide and enhance support for families in both the early and late stages of bereavement [28]. The need for early access to resources and referrals for families has also been reported in recent studies [25,29] and constitutes a critical issue for future targeted postvention interventions.

Our findings offer new insights on how families experience suicide bereavement and may have critical implications for daily practice and future research. Indeed, clinicians should carefully assess the familial processes occurring within the family when counselling individuals bereaved by suicide in order to assess the positive and negative factors shaping their personal experience of bereavement. Moreover, as familial interactions can be significantly impaired in the hours and days after the death by suicide of their relative, implementing early interventions for bereaved families could prevent the onset of deleterious relational and interactional processes within families and negative and psychiatric outcomes among family members. Notably, facilitating and strengthening the support provided within families can be helpful for family members.

With regard to the research agenda, quantitative studies, through large surveys and cohorts, are needed to collect more reliable and meaningful results in samples from multiple cultural and socioeconomic contexts. Notably, our exploratory results indicate the need to confirm or refute their relevance in larger samples and better evaluate the prevalence of the reported processes among families. The association between familial processes and psychiatric outcomes, including prolonged grief disorder, suicidal behaviors, post-traumatic stress disorder or major depressive disorder, should also be assessed. Furthermore, assessing specific processes such as the forgotten grief of siblings or the conflicts between bereaved partners and their in-laws in dedicated studies would be beneficial. Finally, studies comparing the impact within families in diverse situations of bereavement (suicide, violent death or homicide) should also be conducted.

### 4.3. Strengths and Limitations

Our study has several strengths. First, we performed the first well-designed qualitative study to assess the impact of suicide on families and family members, and we included an appropriate sample to reach saturation. Second, we limited the source of bias by employing triangulation and saturation. Third, our results may be of great interest to clinicians or volunteers working with people who have been bereaved by suicide, as they offer relevant insights into the interactions between familial and individual processes in the context of suicide bereavement with respect to the external adversity and help perceived by families and their specific needs to ensure posttraumatic growth. This study offers interesting insights for both daily clinical practice and future research.

However, our study faces several limitations. First, the qualitative design of our study does not allow us to make any definitive causal inferences. Moreover, we assessed the impact of suicide bereavement within families in a small sample of participants. As previously mentioned, our exploratory results must be reinforced by reference to both qualitative and quantitative data and through the use of larger samples. Second, we included individuals rather than families as a whole in our study. A perception bias pertaining to family processes may have occurred, thus impeding the reliability and generalizability of our results. Moreover, we included a diverse sample (parents, spouses, children…) who experienced a suicide in their family 2 to 27 years ago, which raises questions about what types of experience were captured and whether a single qualitative synthesis was relevant to meaningfully integrate these experiences. However, the choice to include diverse individuals rather than families was made to avoid other sources of bias in our study. The inclusion of families may, for example, have introduced barriers to the expression of individual experiences of the family processes for some family members. Third, most of our participants were women, so the generalizability of our results may be limited. However, the gender ratio observed in our study is consistent with those seen in other studies and can be explained by both the greater ability of women to express their feelings in research and the higher prevalence of death by suicide among men.

## 5. Conclusions

We reported that suicide bereavement significantly impacts internal familial interactions via complex emotional and communication processes. We also found that familial coping strategies can shape individual reactions to the death on the part of each member of the family. Family conflicts, taboos or cohesion can occur in the aftermath of such a death. Additional qualitative and quantitative studies are needed to improve our understanding of the ways in which suicide bereavement affects familial interactions and family members.

## Figures and Tables

**Table 1 ijerph-19-13070-t001:** Thematic analysis issued from Sibeoni et al. [11].

Stages	Activities	Rationale	Actions
**Stage 1**	Repeatedly read each interview, as a whole	Obtain a global picture of the interview and become familiar with the interviewee’s verbal style and vocabulary Each new reading of the transcript might also provide new perspectives	Reading the 16 interviews
**Stage 2**	Code the transcript by making notes corresponding to the fundamental units of meanings	Pay particular attention to linguistic details and the vocabulary used by the participant, for instance when he/she uses a metaphor to explain or name a phenomenon, in order to make inductive descriptive notes using the participant’s own words	Exhaustively coding the interviews through the NVivo software to identify all the units of meaning reported by the 16 participants
**Stage 3**	Make conceptual notes through processes of condensation, abstraction, and comparison of the initial notes	Categorize initial notes and reach a higher level of abstraction	Identifying subthemes by compiling the codes collected during Stage 2 through the NVivo software
**Stage 4**	Identify initial themes Provide text quotes that illustrate the main ideas of each theme	Themes are labels that summarize the essence of a number of related conceptual notes. They are used to capture the experience of the phenomenon under study	Identifying general themes by compiling the subthemes that were created during Stage 3, through the NVivo software
**Stage 5**	Identify recurrent themes across transcripts and produce a coherent ordered table of the themes and sub-themes	Move from the particular to the shared across multiple experiences Recurrent themes reflect a shared understanding of the phenomena among all participants During this more analytic stage, researchers try to make sense of the associations between the themes found	Producing a narrative synthesis through a coherent table and narrative of themes and subthemes using the investigator triangulation Making sense of all the units of meaning and of the diverse experiences collected throughout the interviews

**Table 2 ijerph-19-13070-t002:** Characteristics of the participants and of their relatives who died by suicide.

			Participants					Relatives	
	Gender	Age	Relation with the Deceased	Duration of Bereavement (Years)	Recruitment	Gender	Age at Death	Mean of Suicide	Family Type
P1	F	62	Sister	27	Social media	M	37	Firearm	Nuclear
P2	F	48	Mother	1	Social media	M	24	Firearm	Separated
P3	F	59	Sister	14	Social media	M	29	Defenestration	Nuclear (deceased father)
P4	F	48	Sister	24	Social media	M	24	Self-poisoning	Nuclear
P5	F	59	Mother	7	CSP	M	31	Hanging	Separated
P6	M	53	Father	11	Charity	F	15	Hanging	Nuclear
P7	M	54	Father	5	Charity	M	21	Railway	Nuclear
P8	M	63	Father	15	Charity	F	22	Defenestration	Nuclear
P9	F	73	Spouse	11	CSP	M	62	Firearm	Stepfamily
P10	F	41	Daughter	2	Social media	M	70	Unknown	Separated
P11	F	46	Spouse	1.5	CSP	M	44	Phlebotomy	Nuclear
P12	F	38	Daughter	17	Social Media	F	44	Self-poisoning	Nuclear
P13	F	61	Mother	2	CSP	M	30	Defenestration	Separated (deceased father)
P14	F	35	Spouse	2	CSP	M	33	Hanging	Nuclear
P15	F	84	Mother	21	Charity	M	22	Firearm	Nuclear
P16	M	78	Father	7	Charity	M	35	Hanging	Nuclear

CSP = Center for Suicide Prevention.

**Table 3 ijerph-19-13070-t003:** Number of participants supporting each theme.

Themes/Subthemes	Number of Participants (Total = 16)
**Familia trauma**	12
**External adversity**	
Police inquiry Management logistics after the death Material difficulties Social stigma	2 4 2 5
**Familial interactions and individual bereavement**	
Different reactions and coping strategies within the family Climate of «blame» Climate of «fear»	8 6 9
**Communicational and relational processes within the family**	
Families’ difficulties to communicate during bereavement Families’ relational processes during bereavement Siblings as forgotten victims	9 5 6
**Perceived help and support within the family**	
Families’ cohesion External support A lack of support for bereaved families	6 16 8
**Evolution over time**	
Family members’ appeasement Families’ appeasement Family members’ growth Families’ growth	8 5 6 2

## Data Availability

The data that support the findings of this study are available from the corresponding author, EL, upon reasonable request.

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
