# Peer review of "Lived Experiences of Suicide Bereavement within Families: A Qualitative Study"

_ijerph, 2022, doi:10.3390/ijerph192013070_

Round 1
Reviewer 1 Report
This is a well designed qualitative study and the authors take care to follow best practices and to explain their methodology. The presentation of results is also well done. The quotes are illustrative and appropriate context is included. One recommendation for the results is to include some numerical information, such as how many respondents' comments supported a particular theme. A chart with the most significant themes and numerical counts would be helpful. For example, "familial trauma" is the theme listed first in the results. Why is that listed first? Was it the most common theme that emerged? Some explanation would be helpful.
The larger issue to address here is discussion of the significance of the research and the significance of the findings. The authors note that there is scarce literature on suicide bereavement in the family. They write, "A better understanding of the impact of suicide on the emotional, communicational and interactional processes that occur within a family is thus necessary. In 2008, Cerel, Jordan & Duberstein [8] recommended that both qualitative and quantitative studies be conducted to examine the impact of suicide bereavement on the family. However, no well-designed and high quality qualitative study has been conducted since this call to action." This should be followed by a section explaining WHY understanding the impact of suicide bereavement on the family is crucial. The authors could link it to the discussion earlier in the introduction on the negative outcomes related to suicide bereavement. There is only one sentence on familial bonds in determining bereavement outcomes, and that is simply tacked on to evidence showing the importance of social support more generally. Th authors should expand the introduction to clearly explain how understanding familial bereavement following suicide could reduce negative outcomes.
Then, in the discussion, the authors should return to that issue and explain how the study results can be used to inform other research (particularly quantitative research) and how, more specifically, they can be used "in daily clinical practice."
Author Response
- This is a well designed qualitative study and the authors take care to follow best practices and to explain their methodology. The presentation of results is also well done. The quotes are illustrative and appropriate context is included.
We would like to thank the reviewer for her/his positive feedback on our work.
- One recommendation for the results is to include some numerical information, such as how many respondents' comments supported a particular theme. A chart with the most significant themes and numerical counts would be helpful. For example, "familial trauma" is the theme listed first in the results. Why is that listed first? Was it the most common theme that emerged? Some explanation would be helpful.
We added the information on the number of respondents supporting each theme in an additional table (Table 3).
“Familial trauma” was first listed as it was reported by the majority of participants (12 on 16) and was generally reported as the first response to bereavement within families during the interviews.
- The larger issue to address here is discussion of the significance of the research and the significance of the findings. The authors note that there is scarce literature on suicide bereavement in the family. They write, "A better understanding of the impact of suicide on the emotional, communicational and interactional processes that occur within a family is thus necessary. In 2008, Cerel, Jordan & Duberstein [8] recommended that both qualitative and quantitative studies be conducted to examine the impact of suicide bereavement on the family. However, no well-designed and high quality qualitative study has been conducted since this call to action." This should be followed by a section explaining WHY understanding the impact of suicide bereavement on the family is crucial. The authors could link it to the discussion earlier in the introduction on the negative outcomes related to suicide bereavement. There is only one sentence on familial bonds in determining bereavement outcomes, and that is simply tacked on to evidence showing the importance of social support more generally. The authors should expand the introduction to clearly explain how understanding familial bereavement following suicide could reduce negative outcomes.
We do agree that our introduction insufficiently addressed this issue. We thus added a paragraph on the importance of understanding and assessing the impact of suicide bereavement in the family:
“Understanding the impact of suicide bereavement within families is crucial as familial bonds constitute the first social support for many people bereaved by suicide [8]. The lack of familial support could therefore have significant and critical impacts on family members, especially regarding the negative psychological and psychiatric outcomes previously mentioned. Thus, assessing how the communicational and relational processes within the family shape individual bereavement could offer important cues to prevent negative outcomes and favor posttraumatic growth in family members. In the same time, individual processes and negative outcomes in members of the family may also shape the interactions within the family by increasing overall familial burden. Results from studies on this issue could thus inform how to support families and their members in coping with both individual and familial processes of suicide bereavement.” (lines 111 to 120)
- Then, in the discussion, the authors should return to that issue and explain how the study results can be used to inform other research (particularly quantitative research) and how, more specifically, they can be used "in daily clinical practice."
We added a paragraph on the implications of our results in daily practice and for research agenda:
“Our findings offer new insights on how families experience suicide bereavement, which may have critical implications for daily practice and future research. Indeed, clinicians should carefully assess the familial processes occurring within the family when counselling individuals bereaved by suicide in order to assess the positive and negative factors shaping their personal experience of bereavement. Moreover, as familial interactions can be significantly impaired in the hours and days after the death by suicide of their relative, implementing early interventions for bereaved families could prevent the onset of deleterious relational and interactional processes within families and negative and psychiatric outcomes among family members. Notably, facilitating and strengthening the support provided within families can be helpful for family members.
With regard to the research agenda, quantitative studies, through large surveys and cohorts, are needed to collect more reliable and meaningful results in samples from multiple cultural and socioeconomic contexts. Notably, our exploratory results indicate the need to confirm or refute their relevance in larger samples and better evaluate the prevalence of the reported processes among families. The association between familial processes and psychiatric outcomes, including prolonged grief disorder, suicidal behaviors, post-traumatic stress disorder or major depressive disorder, should also be assessed. Furthermore, specific processes such as the forgotten grief of siblings or the conflicts between bereaved partners and their in-laws would benefit to be assessed in dedicated studies. Finally, studies comparing the impact within families in diverse situations of bereavement (suicide, violent death, homicide) should also be conducted.” (lines 517 to 537)
Reviewer 2 Report
This study has several important, albeit understandable, limitations. Notably, the sample size was small for a qualitative study. While the authors note that no new themes emerged in the final two interviews, it is unclear whether this was really a robust enough indicator for saturation. Individual variations in interviews are common and some participants elicit more novelty than others. As such, two interviews with few new themes may reflect saturation or just be two less informative interviews.
The sample is also notably diverse -- parents, spouses, children who experienced a familial suicide 2-27 years ago. The sample was also disproportionately male. These characteristics raise questions about what types of experience are being described and whether a single qualitative synthesis can meaningfully integrate or describe these experiences.
The data analysis is also a bit difficult to understand. It seems perhaps a hybrid of narrative synthesis and thematic analysis? It is helpful to embed examples of what they were doing at each stage -- perhaps in table 1. In particular, understanding how authors synthesized across the diverse experiences and accounted for their diverse audience would be helpful.
Despite the small sample size and diversity of the sample (e.g., duration of bereavement), this study is fairly novel and interesting. It documents important and relevant features of the experiences of people who have experienced suicides in their families. The limitations are largely identified. The background and discussion are articulate and relevant. The quotes are mostly compelling and support the claims.
Author Response
- This study has several important, albeit understandable, limitations. Notably, the sample size was small for a qualitative study.
We do agree and addressed this issue in the limitation section of our article.
“First, the qualitative design of our study does not allow us to make any definitive causal inferences. Moreover, we assessed the impact of suicide bereavement within families in a small sample of participants. As previously mentioned, our exploratory results must be reinforced by reference to both qualitative and quantitative data and through the use of larger samples.” (lines 548 to 552)
- While the authors note that no new themes emerged in the final two interviews, it is unclear whether this was really a robust enough indicator for saturation. Individual variations in interviews are common and some participants elicit more novelty than others. As such, two interviews with few new themes may reflect saturation or just be two less informative interviews.
We do agree that the way we reached saturation was insufficiently described. We added precision on this issue:
“According to the taxonomy developed by Saunders et al. (2018) [17], the following four types of saturation are defined: (a) theoretical saturation (i.e., the development of theoretical categories), (b) inductive thematic saturation (i.e., the emergence of new codes or themes), (c) a priori thematic saturation (i.e., the degree to which identified codes or themes are exemplified in the data), and (d) data saturation (i.e., the degree to which new data repeat what was expressed in previously collected data). Given the homogeneity of the sample (i.e., French family members bereaved by suicide), a total of 15 to 20 participants were expected to be necessary to reach the point of saturation. The study was concluding after 16 participants had been included, as the four types of saturation were reached. Indeed, no new codes nor new themes emerged during the final two interviews (theoretical and inductive thematic saturation) and the data collected during these two interviews repeated what was previously reported by the other participants (data saturation). Moreover, a priori thematic saturation was also reached as the codes and themes that we identified were highly exemplified in the data collected during all the interviews.” (lines 187 to 200)
- The sample is also notably diverse -- parents, spouses, children who experienced a familial suicide 2-27 years ago. The sample was also disproportionately male. These characteristics raise questions about what types of experience are being described and whether a single qualitative synthesis can meaningfully integrate or describe these experiences.
We do agree with the reviewer and addressed this issue in the limitation section of our article:
“Moreover, we included a diverse sample (parents, spouses, children…) who experienced a suicide in their family 2 to 27 years ago, which raises questions about what types of experience were captured and whether a single qualitative synthesis was relevant to meaningfully integrate these experiences. However, the choice to include diverse individuals rather than families was made to avoid other sources of bias in our study.” (lines 554 to 559)
- The data analysis is also a bit difficult to understand. It seems perhaps a hybrid of narrative synthesis and thematic analysis? It is helpful to embed examples of what they were doing at each stage -- perhaps in table 1. In particular, understanding how authors synthesized across the diverse experiences and accounted for their diverse audience would be helpful.
We do confirm that we used a hybrid of thematic and narrative synthesis. We clarified this issue in the Table 1.
- Despite the small sample size and diversity of the sample (e.g., duration of bereavement), this study is fairly novel and interesting. It documents important and relevant features of the experiences of people who have experienced suicides in their families. The limitations are largely identified. The background and discussion are articulate and relevant. The quotes are mostly compelling and support the claims.
We would like to thank the reviewer for her/his positive feedback on our work.